# A Systematic Review of MicroRNAs Involved in Cervical Cancer Progression

**DOI:** 10.3390/cells10030668

**Published:** 2021-03-17

**Authors:** Rhafaela Lima Causin, Ana Julia Aguiar de Freitas, Cassio Murilo Trovo Hidalgo Filho, Ricardo dos Reis, Rui Manuel Reis, Márcia Maria Chiquitelli Marques

**Affiliations:** 1Molecular Oncology Research Center, Barretos Cancer Hospital, Teaching and Research Institute, Barretos-SP 14784-400, Brazil; rhafaela-lima@hotmail.com (R.L.C.); aaguiardefreitas@gmail.com (A.J.A.d.F.); ruireis.hcb@gmail.com (R.M.R.); 2Instituto do Câncer do Estado de São Paulo (ICESP), São Paulo-SP 01246-000, Brazil; muriloth@hotmail.com; 3Gynecologic Oncology Department, Barretos Cancer Hospital, Barretos, São Paulo 14784-400, Brazil; drricardoreis@gmail.com; 4Life and Health Sciences Research Institute (ICVS), Medical School, University of Minho, 4704-553 Braga, Portugal; 5ICVS/3B’s-PT Government Associate Laboratory, 4806-909 Braga/Guimarães, Portugal; 6Barretos School of Health Sciences, Dr. Paulo Prata–FACISB, Barretos, São Paulo 14785-002, Brazil

**Keywords:** cervical cancer, microRNA, cervical cancer progression

## Abstract

To obtain a better understanding on the role of microRNAs in the progression of cervical cancer, a systematic review was performed to analyze cervical cancer microRNA studies. We provide an overview of the studies investigating microRNA expression in relation to cervical cancer (CC) progression, highlighting their common outcomes and target gene interactions according to the regulatory pathways. To achieve this, we systematically searched through PubMed MEDLINE, EMBASE, and Google Scholar for all articles between April 2010 and April 2020, in accordance with the PICO acronym (participants, interventions, comparisons, outcomes). From 27 published reports, totaling 1721 cases and 1361 noncancerous control tissue samples, 26 differentially expressed microRNAs (DEmiRNAs) were identified in different International Federation of Gynecology and Obstetrics (FIGO) stages of cervical cancer development. It was identified that some of the dysregulated microRNAs were associated with specific stages of cervical cancer development. The results indicated that DEmiRNAs in different stages of cervical cancer were functionally involved in several key hallmarks of cancer, such as evading growth suppressors, enabling replicative immortality, activation of invasion and metastasis, resisting cell death, and sustained proliferative signaling. These dysregulated microRNAs could play an important role in cervical cancer’s development. Some of the stage-specific microRNAs can also be used as biomarkers for cancer classification and monitoring the progression of cervical cancer.

## 1. Introduction

Cervical cancer (CC) is the most important human papillomavirus (HPV)-induced disease, being a major worldwide public health problem [1,2]. It corresponds to the fourth most common cancer among women in the world, with 569,847 new cases and 311,365 deaths registered annually [3]. There is an estimation that by 2030 the number of cervical cancer cases in the world will increase by around 50% [4]. In Brazil, it is estimated that there were 16,370 new cases for each year of the 2018–2019 biennium, with an estimated risk of 15.43 cases for every 100,000 women [5].

Cervical cancer morbidity and mortality have declined in the last 40 years in many countries, mainly due to the effectiveness of the widespread implementation of prevention programs [6]. Effective screening programs have been introduced in many industrialized countries. However, in low and middle-income countries (LMICs), “Pap smear” programs have mainly been limited to offering the test to women present in primary health care centers and other health clinics [7,8]. For this reason, the current prevention programs in these countries are not having a major impact in terms of decreasing mortality [9]. In addition, the survival rate in a late stage is approximately 15% [10,11]. Therefore, the prognosis and 5-year survival rate of patients with advanced cervical cancer still remain poor. The standard treatment for patients diagnosed with locally advanced stages (LACC) consists of radiotherapy in combination with cisplatin-based chemotherapy [12,13]. However, approximately 50% of patients do not respond to standard treatment; therefore, these patients have a higher rate of recurrence and a lower survival rate in the first five years [14].

The fundamental step in the malignant transformation of cervical cells is the interaction between these cells and HPV [15,16]. This infection can originate pre-neoplastic lesions [17], known as cervical intraepithelial neoplasia (CIN) (Figure 1). Based on their association with cervical cancer and precursor lesions, HPVs can also be grouped into high-risk and low-risk HPV types. Low-risk HPV types are those that cause genital warts but are not carcinogenic, including types 6, 11, 42, 43, and 44 [16,18]. High-risk HPV types are considered carcinogenic, and include 16, 18, 31, 33, 34, 35, 39, 45, 51, 52, 56, 58, 59, 66, 68, and 70 [6,16].

Wentzensen et al. [19] confirmed that HPV integration sites are randomly distributed over the whole genome, with a clear predilection for fragile sites. Thereby, this interaction is more frequent in chromosomal fragile sites (CFS); these regions of the chromosome are the most prone to genetic and epigenetic changes [15,20]. It is known that approximately 50% of microRNAs (miRNAs) are located in these fragile sites or in regions associated with cancer [21], and although the function of miRNAs as oncogenes or as tumor suppressors has already been well characterized [22], the mechanisms of action of miRNAs in the progression of CC have been studied only recently [23].

miRNAs are small non-coding RNAs (19 to 24 nucleotides) that regulate the expression of coding genes [24] with crucial roles in important biological processes, such as proliferation, cell growth, and apoptosis, among others [25,26]. The canonical pathway of biogenesis of miRNAs begins in the nucleus after the transcription of the miRNA by polymerase II—the synthesis of the primary miRNA (Pri-miRNA). Then, the Drosha enzyme linked to the DGCR8 cofactor cleaves the end of the poly AAA tail, giving rise to the precursor miRNA (Pre-miRNA), which is exported to the cytoplasm via exportin 5. In the cytoplasm, the Pre-miRNA clamp is cleaved by Dicer enzyme linked to the TRBP cofactor resulting in a duplex miRNA, at which point the RISC complex guides only one of the strands of mature miRNA to the target mRNA. Then, the interaction in the 3’UTR region occurs [27]. The result of the interaction between the mRNA and the miRNA can lead to inhibition of translation, in the event of imperfect pairing, or the degradation of the mRNA, which occurs in cases of perfect pairing between the miRNA and the mRNA [28] (Figure 2).

Several factors are necessary for the development of CC, including the interaction of viral and environmental factors and the host’s immune capacity—steps that trigger tumor growth, invasion, and metastasis [29]. In addition, evidence about epigenetic regulatory mechanisms has steadily increased over the past two decades and has focused on the dysregulation of oncogenes and tumor suppressor genes as the main generator of the malignant phenotype [30]. In this sense, miRNAs play an important role as regulators of cellular processes, which involve apoptosis, cell cycle progression, metastases, invasion, and chemoradioresistance [31,32]. In an effort to understand the important roles of various miRNAs during cervical tumor genesis, many studies have been carried out to describe the real regulatory role of these molecules.

miRNAs can be subdivided into oncogenic miRNAs (oncomiRs) and tumor suppressor miRNAs (tsmiRs) [33]. For a future in personalized medicine for the treatment of CC, in which we can use miRNAs as possible therapeutic targets, we first need to understand the oncogenic or tumor suppressive role of miRNAs and how their regulation may affect CC development and progression. Blocking and downregulating oncomiRs may play an important role in the treatment of this tumor, while the overexpression of tsmiRs may provide anti-cancer therapeutic effects [22,34,35,36]. On the other hand, different tsmiRs have different mechanisms for suppressing CC [25,37]. Most of them, such as miR-1284, miR-573, miR-433, miR-424-5p, and miR-361-5p, can affect cellular proliferation, migration, and apoptosis. In the present review, we conducted an extensive search of the literature, which yielded information concerning 28 miRNAs associated with the progression of CC, as identified in 27 studies.

## 2. Materials and Methods

The Preferred Reporting Items for Systematic Reviews and Meta-Analyses (PRISMA) tool was used to ensure transparent reporting [38,39]. Both reviewers (RLC and AJAdF) performed an independent assessment of the quality of all included studies. Discrepancies were discussed and resolved by consensus.

### 2.1. Search

Potential articles for inclusion in the current study were searched for through PubMed MEDLINE, Embase, and Google Scholar databases between April 2010 and April 2020 using combinations of the keywords using Medical Subject Headings (MeSH) terms: “uterine cervix tumor” or “cervical intraepithelial neoplasia 2” or “cervical intraepithelial neoplasia 3,” and “miRNA” or “microRNA,” and “progression disease.” The search was followed by checking references listed in the identified articles for potentially eligible original reports. After merging the results of the three searches, all records were individually screened by title and abstract by two authors (RLC and AJAdF). Discrepancies were discussed and resolved by consensus, resulting in either inclusion or exclusion for any full text.

### 2.2. Inclusion and Exclusion Criteria

We employed the PICO acronym (participants, interventions, comparisons, outcomes) for predetermined criteria for inclusion [38,40]; the criteria can be seen in Table 1. All miRNAs involved with cervical cancer studies were used in the present study if they met the following criteria: (i) original experimental report to analyze miRNA profile between human cancerous and noncancerous cervical tissues and/or CIN; (ii) articles published in the last 10 years; (iii) original articles published in either English or Spanish; and (iv) impact factor greater than or equal to 2. The exclusion criteria were as follows: (i) articles entirely in silico; (ii) an article that belongs to a meta-analysis, thesis, series, abstract, comment, review, letter, or editorial; (iii) duplicate reports; (iv) studies that did not have control tissues and studies that did not use human samples; (v) articles that did not use cell lines and patients’ samples; (vi) studies that did not to validate the interaction between the target gene and miRNA.

### 2.3. Data Extraction

Data were extracted from all publications that met the inclusion criteria according to two independent reviewers using a standardized data extraction form. We extracted the following information for each eligible study; study design, year of publication, specimens (cell lines and patients sample type), number of patients, International Federation of Gynecology and Obstetrics (FIGO)stages, miRNA deregulated, target interaction, target interaction validation, hallmarks of cancer, and clinical outcome. In case data were unavailable in the manuscript or supplementary files, the articles were excluded.

## 3. Results

Hereafter, we selected 113 studies for a full-text review, and 27 studies [41,42,43,44,45,46,47,48,49,50,51,52,53,54,55,56,57,58,59,60,61,62,63,64,65,66,67] were eventually identified as eligible and were included in the present study (Figure 3). All 27 articles included were observational and experimental studies, 11 of which were carried out with experiments on animals. Included patient numbers ranged from 21 to 147, and resulted in a total of 1785 patients (1721 CC cases and 1361 noncancerous control tissue cases).

### 3.1. Study Population

All samples used in the included studies evaluated human tissues; the vast majority of normal tissues consisted of normal tissues adjacent to the tumor; three studies included healthy tissues, and the other three used CIN samples. All studies included patients according to FIGO stages. We also removed circulating miRNA studies with serum or plasma samples due to the lack of a sufficient number of independent studies. The main characteristics extracted from various datasets are summarized in Table 2, which include the number of patients investigated, the measurement platform, the pathological description of samples, and the identities of differentially expressed miRNAs (DEmiRNAs).

### 3.2. miRNAs Involved in Cervical Cancer Progression

Interestingly, all DEmiRNAs was confirmed by qRT-PCR. We identified 26 miRNAs reflecting the progression of cervical cancer. There are 19 downregulated (miR-1284, miR-573, miR-433, miR-424-5p, miR-361-5p, miR-383-5p, miR-335-5p, miR-874, miR-132, miR-411, miR-337-3p, miR-3941, miR-545, miR-143, miR-107, miR-1, miR-139-3p, miR-195, and 2861) and seven upregulated (miR-96-5p, miR-199b-5p, miR-93, miR-200a, miR-224, miR-92a, and miR-31) miRNAs among different stages of cervical cancer compared to normal tissue. These data are summarized in Table 3.

Hanahan and Weinberg [76] described ten hallmarks of cancer, of which five have the most impact on miRNA regulation and CC progression in according with this search. In this review, we focus on the following: proliferation, enabling replicative immortality, activation of invasion and migration (metastasis), resisting cell death (apoptosis), and sustained proliferative signaling in CC. In fact, we identified different miRNAs involved in these processes and responsible for the CC progression (Figure 4). In this review, we summarized data from an extensive bibliographical investigation regarding miRNAs associated with clinical outcome by overall survival (OS) (Table 3).

### 3.3. Targets of miRNAs Involved in the Cervical Cancer Progression

The conventional treatment for patients diagnosed with LACC is radiotherapy concomitant with cisplatin [12]. However, approximately 50% of LACC patients receiving conventional treatment exhibit recurrence [77] or poor prognosis in the first five years of disease [14]. Thus, resistance to therapy is a major obstacle for efficient cervical cancer treatment [78]. Recently, several miRNAs have been associated with the survival and prognosis of CC patients [79]. Consequently, the expression and regulation of their targets have become molecular markers with clinical relevance.

For instance, the expression profiles of 26 cancer-related miRNAs from 57 CC tumor tissue were analyzed. Through prediction target tools, the authors identified the target interaction between DEmiRNAs and target genes. Finally, we associated this interaction of tsmiRs and oncomiRs with hallmarks of cancer (Figure 5), specifically, pathways related to CC progression. The physical interactions between miRNAs and target genes predicted by in silico tools were validated using luciferase assay; these data are summarized in Table 4.

## 4. Discussion

In summary, the expression levels of several miRNAs were repeatedly found to be associated with progression towards CC by regulating target genes in different studies [45,46,47,53,80]. In this sense, many DEmiRNAs and target genes have been studied, the majority of which are associated with apoptosis, cell cycle control, migration, genetic instability, cell adhesion, and metastasis. In this study, we identified miRNAs responsible for regulating different genes that act in CC progression pathways.

Most of the dysregulated miRNAs were downregulated (miR-1284, miR-573, miR-433, miR-424-5p, miR-361-5p, miR-383-5p, miR-335-5p, miR-874, miR -132, miR-411, miR-337-3p, miR-3941, miR-545, miR-143, miR-107, miR-1, miR-139-3p, miR-195, and miR-2861), and low expression of them was associated with a poor prognosis. Huang et al [61]. identified that miR-139-3p is capable of suppressing cell proliferation, migration, and invasion, and induces apoptosis by negative regulation of NOB1 in CC cells. These results suggest that miR-139-3p can act as a tumor suppressor, suppressing NOB1 expression in the progression of CC. Another study [66] suggested that miR-424 contributes to the progression of cervical cancer at least partly via upregulation of target gene Chk1 expression and phosphorylation of Chk1 protein. miR-424 might become a potential predictor for prognosis and a candidate target therapy for cervical cancer patients. In addition, a study [67] demonstrated that the interaction of the E6 oncoprotein could also cause a decrease in the expression of a tsmiR, by which HPV16 E6 is able to decrease the level of miR-2861 expression, resulting in the suppression of the EGFR/AKT2/CCND1 pathway in cervical cancer cells. In summary, a new regulatory network was found, employing HPV16 E6, miR-2861, and the EGFR/AKT2/CCND1 signaling pathway to syntonize proliferation, apoptosis, and invasion in cervical cancer cells. This understanding of the single molecular pathway, HPV16 E6/miR-2861/EGFR/AKT2/CCND1, may provide a new insight into exploring additional strategies for cervical cancer therapy in the future.

On the other hand, our study identified seven upregulated miRNAs (miR-96-5p, miR-199b-5p, miR-93, miR-200a, miR-224, miR-92a, and miR-31), among which it was seen that such upregulated miR-93 was correlated with shorter overall survival than low expression of miR-93 (P < 0.01). Patients with low expression levels of CDKN1A mRNA more often had shorter overall survival than those with high expression levels of CDKN1A mRNA (P < 0.01) [55]. In addition, the survival rate of patients with upregulated miR-93 downregulated CDKN1A was the shortest (P < 0.01). In another study [63], it was reported that miR-92a, which is upregulated in cervical cancer, plays an oncogenic role in cervical cancer cell proliferation by directly targeting p21, and thus promoting cell cycle progression. Therefore, these findings suggest that miR-92a can constitute a useful therapeutic target for cervical cancer. Wang et al. [65], demonstrated that the overexpression of miR-31 was associated with poor prognosis and aggressive phenotype of cervical cancer. These authors also implied that miR-31 plays an important role in the regulation of cervical cancer’s malignant behavior, including cell proliferation and invasion, by directly targeting ARID1A. In general, miR-31 provides new insights into prognostic diagnosis and therapeutic strategies for patients with cervical cancer.

Multiple miRNAs have been identified by regulating different genes from a regulatory axis between long and circular non-coding RNAs, miRNAs, and target genes [43,45,52]. For example, functional assays have shown that inhibition of SNHG12 expression decreased cell proliferation both in vitro and in vivo, indicating that SNHG12 plays a critical role in the progression of cervical cancer [81]. MiR-424-5p was downregulated in cervical cancer tissues, and in addition, there was a negative correlation between SNHG12 and the expression of miR-424-5p in cervical cancer tissues [81]. The silencing of miR-424-5p in CC cells with SNHG12 depletion reversed the effects on apoptosis, migration, and cell invasion induced by the SNHG12 knockdown. Another study [80], showed that TTN-AS1 positively modulates E2F3 expression by regulating miR-573 in CC cells. These data revealed that the TTN-AS1 lncRNA was involved in the progression of CC cells by regulating the miR-573-E2F3 axis. Ding and Zhang [43] demonstrated that the circular RNA circ-ATP8A2 promoted the progression of CC cells regulating the signaling of the miR-433/EGFR axis.

Currently, treatment for metastatic or recurrent CC is still a challenge, since the overall prognosis for this disease in advanced stages remains poor. Bearing in mind that some miRNAs are regulators of tumorigenesis in some types of cancer, and therefore represent powerful therapeutic targets. Thus, the incorporation of new therapeutic targets could be more effective for the treatment of this advanced CC. Coeh et al. [13] described new immunotherapeutic approaches that may demonstrate promising results in the future. In this sense, several target therapies have been explored; an example of this is the BEATcc study (NCT03556839), which is a randomized phase 3 study that uses standard platinum-based chemotherapy plus paclitaxel with bevacizumab versus the ligand-programmed cell death inhibitor 1 (PD-L1) atezolizumab compared to platinum chemotherapy plus paclitaxel and bevacizumab in metastatic, persistent, or recurrent CC. To date, the study is still in the recruitment of patients, and the test is expected to close in the third quarter of 2022, and it is hoped that mature BEATcc test data can be reported by 2023 [82].

Forterre et al. [83] highlighted that although the role of miRNA as a regulator of several physiological processes is evident, the main challenge today is in delivering tissue-specific molecules to the cells of interest. Therefore, several approaches have been studied in both in vitro and in vivo assays, in order to evaluate the best methodology for delivering miRNA in tumors for therapeutic purposes. In this sense, CC has many advantages for miRNA therapeutic delivery when compared with other tumors. Since local administration is limited to accessible tumors, as is the case with CC, the benefits of this therapy can be quite interesting, as the intratumoral injection toxicity of mimetizers or miRNA inhibitors is significantly greater than for those treatments administered by systemic routes [83]. Hanna et al. demonstrated in a review stating that there are studies prior to phase 3 to investigate new drug candidates for miRNA drugs, although there is still no phase 3 study registered in the clinictrials.gov database [84].

Another advantage of miRNAs is the potential to be used as biomarkers for accurate diagnosis, targeted treatment, and prediction of response to treatment. This is especially important in CC, a totally heterogeneous disease, in which a multi-marker approach would be preferable [85]. In this context, Xiong et al. [86] identified a multi-marker panel of nine miRNAs for breast carcinoma that has been shown to significantly improve the reliability of the diagnosis of breast cancer. However, for CC there are still no studies that include multi-marker miRNAs. However, research on miRNAs as biomarkers is still in its early stages, so at the moment, the findings generally lack reproducibility.

Among the five hallmarks of cancer [76] identified in this study (Figure 5), the hallmark that showed the most DEmiRNAs was sustained proliferative signaling, with 18 downregulated tsmiRs and six upregulated oncomiRs in patients with CC. Among the different miRNAs identified in this pathway, Ou et al. [49] found that miR-132 can play an important role in regulating the progression of cervical cancer, contributing to the growth and progression of cervical cancer through the depression of the expression of the RDX oncogene as underlying mechanism. On the other hand, it was seen that miR-411 inhibited the progression of cervical cancer when interacting directly with STAT3; the authors also indicated that this miRNA may represent a new potential therapeutic target and prognostic marker for patients with this disease [50]. The enabling replicative immortality marker showed the least miRNAs that regulate genes associated with this pathway. All identified genes that result in promoting replicative immortality are associated with the cell cycle, allowing the cell to migrate early from phase G1 to phase S, as is the case with the CDKN1A [55] and p21 [63] genes that are downregulated by miRNAs miR-93 [55] and miR-92a [63], respectively. Other miRNAs have also been identified in the pathways of evading growth suppressors, activation of invasion and metastasis, and resisting cell death. According to Zhou et al. [64], the interaction axis between miR-195 and Smad3 provides an insight into cervical cancer metastasis and may represent a novel therapeutic target. For this reason, we highlight the importance of identifying pathway regulating miRNAs responsible for the progression of cervical cancer so that potential biomarkers can be identified as future therapeutic targets.

## 5. Conclusions

In conclusion, the evidence highlighted in the present study showed that miRNAs affect various biological pathways associated with cancer development and progression, and clinical outcome and treatment response in CC, reinforcing their roles as key players in carcinogenesis. Consequently, their use in cervical cancer for clinical outcome prediction and therapy improvement may be valuable, facilitating disease classification, monitoring of the progression of cervical cancer, and therapeutic use for patients with resistant treatment disease.

## Figures and Tables

**Figure 1 cells-10-00668-f001:**
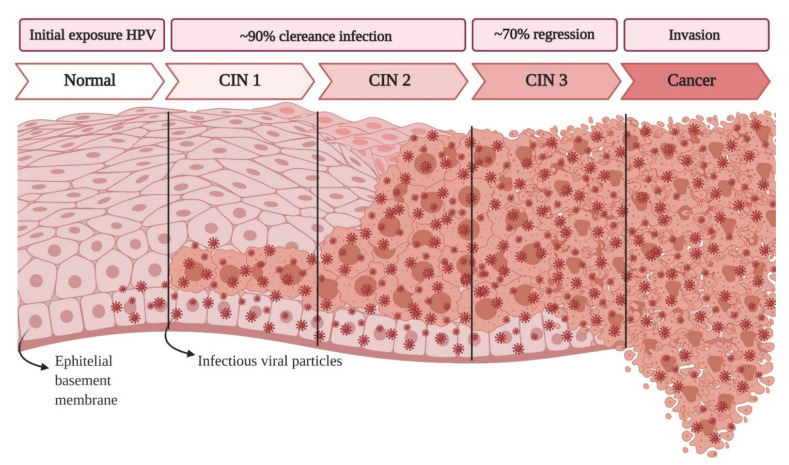
Classification of normal squamous epithelial cells and human papillomavirus (HPV) infections in normal precancerous lesions (cervical intraepithelial neoplasia grades 1, 2, and 3 “CIN 1, CIN 2, and CIN 3”) and cervical cancer.

**Figure 2 cells-10-00668-f002:**
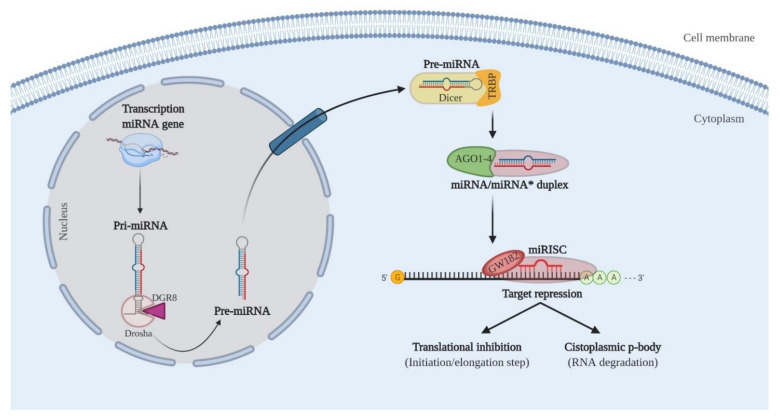
Biogenesis and modulation of microRNA activity.

**Figure 3 cells-10-00668-f003:**
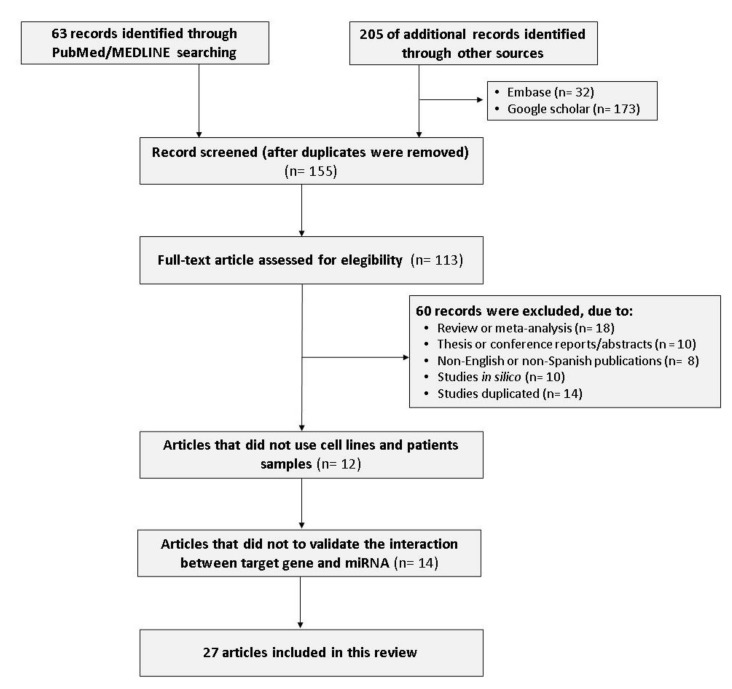
The flow chart of data identification and selection.

**Figure 4 cells-10-00668-f004:**
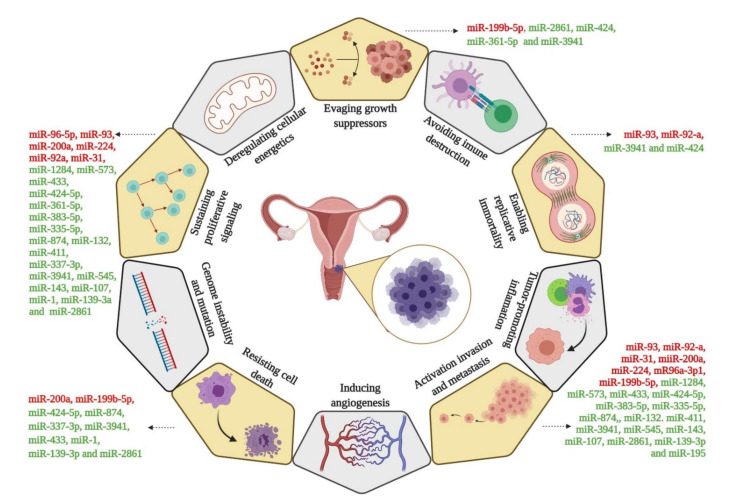
Different miRNAs affecting the hallmarks of cervical cancer.

**Figure 5 cells-10-00668-f005:**
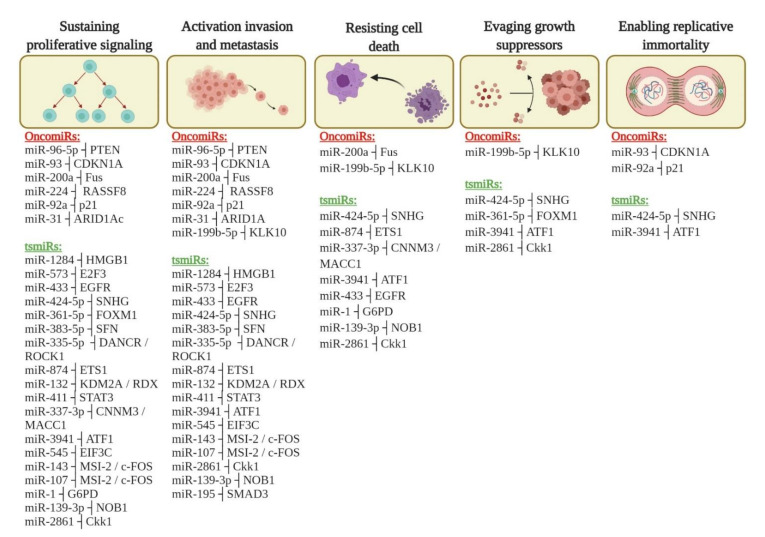
miRNAs involved in post-transcriptional regulatory interactions in cervical cancer. The oncogenic miRNAs (oncomiRs) and tumor suppressor miRNAs (tsmiRs) associated with various hallmark characteristics of cervical cancer are listed under the red and green subheadings, respectively. Each of these miRNAs can post-transcriptionally regulate a large number of genes involved in cervical cancer. The symbol ┤indicates a negative regulating miRNA and target gene.

**Table 1 cells-10-00668-t001:** Eligibility criteria in accordance with PICO (participants, interventions, comparisons, outcomes).

Domain	Inclusion Criteria
**Patients (P)**	Patients with cervical cancer or cervical intraepithelial neoplasia grade 2/3
**Interventions (I)**	Differentially expressed miRNAs
**Comparators (C)**	Non-neoplastic control tissue or cervical cancer FIGO stage I
**Outcomes (O)**	Differentially expressed miRNAs indicate a statistically significant difference in the overall survival of patients

**Table 2 cells-10-00668-t002:** Characteristics of studies included in the current review.

Author, Year	Country	Cell Lines	Sample	N	Non-Neoplastic Control Tissue (n)	Cervical Cancer Samples FIGO Stage (n)
Chen and Li, 2018 [41]	China	MS751, HeLa, SiHa e C33 A and ECS	Tissue	82	NAT (82)	I (28) II (54)
Chen et al., 2018 [42]	China	SiHa, HeLa, C-33A, Me180, Ms751 and Ect1/E6E7	Frozen Tissue	45	NAT (45)	I-II (26) III-IV (19)
Ding and Zhang, 2019 [43]	China	HeLa, SiHa, C-33A, SW75 and HcerEpic	FrozenTissue	46	NAT (46)	I (31) II (15)
Dong et al., 2018 [44]	China	C-33A, ME-180, CaSki, HeLa, SiHa and NC104	Tissue	81	NAT (81)	I (46) II (35)
Gao et al., 2019 [45]	China	SiHa, HeLa, C-33A, Me180 and Ms751 and Ect1/E6E7	Frozen Tissue	66	NAT (66)	I-II (29) III-IV (37)
Hu et al., 2019 [46]	China	HeLa, SiHa, C-33A and Ect1-E6E7	Frozen Tissue	33	NAT (33)	I-II (13) III-IV (20)
Liang et al., 2019 [47]	China	CaSki, SW756, SiHa, C-33A, HeLa, ME-180 and Ect1-E6E7	Frozen Tissue	65	NAT (65)	I-II (37) III-IV (28)
Liao et al., 2018 [48]	China	SiHa, HeLa, C-33A, CaSki and Ect1/E6E7	Frozen Tissue	49	NAT (49)	I-II (23) III-IV (26)
Ou et al., 2019 [49]	China	C-33A, SiHa, and CaSk	Tissue	81	NAT (81)	Ib-IIa (37) IIb-IIIa (44)
Shan et al., 2019 [50]	China	HeLa, SiHa, CaSki, C-33A and Ect1/E6E7	Frozen Tissue	45	NAT (45)	I-II (17) III-IV (28)
Shao et al., 2019 [51]	China	C-33A, MS751, SiHa, HeLa, ME-180, CaSki and NC104	Tissue	37	NAT (37)	I-II (19) III-IV (18)
Wang et al., 2020 [52]	China	HeLa, SiHa, C-33A, SW756 and HcerEpic	Tissue	55	NAT (55)	I (31) II (24)
Xu et al., 2018 [53]	China	HeLa, CaSki, SiHa, ME-180, MS-751, C-33 A, Ect1/E6E7 and HcerEpic	Frozen Tissue	70	NAT (70)	I-II (36) III-IV (34)
Xu et al., 2019 [54]	China	C-33A, SiHa, ME-180, HeLa, CaSki and NC104	Tissue	92	NAT (92)	Ib-IIa (46) IIb-IIIa (46)
Zhang et al., 2018 [55]	China	HeLa	Tissue	100	NAT (100)	I (56) II (32) III (8) IV (4)
Hu et al., 2019 [56]	China	SiHa, CaSki, HeLa, C4-1, and NC104	Frozen Tissue	21	NAT (21)	I (34) II (27)
Zhu et al., 2018 [57]	China	HeLa, CaSki, C4-1, SiHa, and GH329	Tissue	52	NAT (52)	I-II (30) III-IV (23)
Liang et al., 2017 [58]	China	HeLa, C-33A, SiHa, CaSki, and Hct1/E6E7	Frozen Tissue	65	NAT (65)	I-II (24) II-IV (41)
Dong et al., 2017 [59]	Japan	HeLa, SiHa and H8	Frozen Tissue	58	NAT (58)	I-II (38) III-IV (20)
Hu et al., 2016 [60]	China	HeLa, SiHa, C-33A and H8	Frozen Tissue	57	NAT (57)	I (28) II (14) III (9) IV (3)
Huang et al., 2016 [61]	China	HeLa, SiHa, CaSki, C-33A, and HaCaT	Frozen Tissue	40	NAT (40)	I-II (40)
Huang et al, 2016 [62]	China	SiHa and CaSki	Frozen Tissue	190	HT (64)	I (86) II (40)
Su et al., 2017 [63]	China	HeLa	Frozen Tissue	74		I-II (39) III-IV (35)
Zhou et al., 2016 [64]	China	HeLa, SiHa, CaSki, ME-180, C-33A and HaCaT	Tissue	50		IB (27) IC-IV (23)
Wang et al., 2014 [65]	China	HeLa, SiHa, CaSki, ME-180, C-33A and HaCaT	Tissue	27		I (20) II (7)
Xu et al., 2013 [66]	China	SiHa and CaSki	Frozen Tissue	147		I (108) II (39)
Xu et al., 2016 [67]	China	SiHa, CaSki, HEK293T and HaCaT	Frozen Tissue	57	NAT (57)	I (35) II (22)

NAT: normal tissue adjacent to the tumor. HT: healthy tissue.

**Table 3 cells-10-00668-t003:** Differential expression of microRNAs (miRNAs) and targets genes in cervical cancer (CC) samples and cell lines.

miRNA	Target Interation	Target Prediction Tool	Target Validated	REF
miR-1284	HMGB1	miRanda	Luciferase	[41]
miR-573	E2F3	Targetscan	Luciferase	[42]
miR-433	circ-ATP8A2/EGFR	Targetscan	Luciferase	[43,58]
miR-424-5p	SNHG12	StarBase v2.0	Luciferase	[44,66]
miR-361-5p	SBF2-AS/FOXM1	Previous studies [68,69]	Luciferase	[45]
miR-383-5p	LINC01128/SFN	StarBase v2.0	Luciferase	[46]
miR-335-5p	DANCR/ROCK1	DIANA-LncBase V.2; Targetscan Human 7.2	Luciferase	[47]
miR-874	ETS1	TargetScan7.1 and microRNA.org	Luciferase	
miR-132	KDM2A/RDX	PicTar algorithm	Luciferase	[49]
miR-411	STAT3	TargetScan and miRanda	Luciferase	[50]
miR-96-5p	STXBP5-AS1/PTEN	TargetScan and miRanda	Luciferase	[51]
miR-337-3p	hsa_circ_0001038, CNNM3/MACC1	Circular RNA Interactome and TargetScan	Luciferase	[52]
miR-199b-5p	KLK10	TargetScan and miRanda	Luciferase	[53]
miR-3941	lncRNA RP11-552M11.4/ATF1	DIANA tool LncBase v.2 and TargetScan	Luciferase	[54]
miR-93	CDKN1A	Previous studies [70,71,72,73,74]	Luciferase [70,71,72,73]	[55]
miR-545	circ_0067934/EIF3C	Circular RNA Interactome and TargetScan	Luciferase	[56]
miR-200a	XIST/Fus	Starbase	Luciferase	[57]
miR-143	MSI-2/c-FOS	RNA-IP	Luciferase	[59]
miR-107	MSI-2/c-FOS	RNA-IP	Luciferase	[59]
miR-1	G6PD	RNA-IP	Luciferase	[60]
miR-139-3p	NOB1	TargetScan, miRanda, and Diana microT computational algorithms	RIP-Chip and Luciferase	[61]
miR-224	RASSF8	TargetScan Human 7.0	Luciferase	[62]
miR-92a	p21	TargetScan, PicTar and microbase	Luciferase	[63]
miR-195	SMAD3	Previous study [75]	Luciferase [75]	[64]
miR-31	ARID1A	TargetScan, pictar, and miRanda	Luciferase	[65]
miR-2861	EGFR/ AKT2/CCND1	TargetScan, pictar, miRanda and Microcosm Targets	Luciferase	[67]

**Table 4 cells-10-00668-t004:** miRNAs involved in cervical cancer’s clinical outcome.

miRNA	Status	Methodology	Outcome	REF
miR-1284	D	RT-qPCR	Poor prognosis	[41]
miR-573	D	RT-qPCR	Poor prognosis	[42]
miR-433	D	RT-qPCR	Poor prognosis	[43,58]
miR-424-5p	D	RT-qPCR	Poor prognosis	[44,66]
miR-361-5p	D	RT-qPCR	Poor prognosis	[45]
miR-383-5p	D	RT-qPCR	Poor prognosis	[46]
miR-335-5p	D	RT-qPCR	Poor prognosis	[47]
miR-874	D	RT-qPCR	Poor prognosis	
miR-132	D	RT-qPCR	Poor prognosis	[49]
miR-411	D	RT-qPCR	Poor prognosis	[50]
miR-96-5p	U	RT-qPCR	Poor prognosis	[51]
miR-337-3p	D	RT-qPCR	Poor prognosis	[52]
miR-199b-5p	U	RT-qPCR	Poor prognosis	[53]
miR-3941	D	RT-qPCR	Poor prognosis	[54]
miR-93	U	RT-qPCR	Poor prognosis	[55]
miR-545	D	RT-qPCR	Poor prognosis	[56]
miR-200a	U	RT-qPCR	Poor prognosis	[57]
miR-143	D	RT-qPCR	Poor prognosis	[59]
miR-107	D	RT-qPCR	Poor prognosis	[59]
miR-1	D	RT-qPCR	Poor prognosis	[60]
miR-139-3p	D	RT-qPCR	Poor prognosis	[61]
miR-224	U	RT-qPCR	Poor prognosis	[62]
miR-92a	U	RT-qPCR	Poor prognosis	[63]
miR-195	D	RT-qPCR	Poor prognosis	[64]
miR-31	U	RT-qPCR	Poor prognosis	[65]
miR-2861	D	Microarray and RT-qPCR	Poor prognosis	[67]

D: dowregulated. U: upregulated.

## Data Availability

No new data were created or analyzed in this study. Data sharing is not applicable to this article.

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
