# Peer review of "A Systematic Review of MicroRNAs Involved in Cervical Cancer Progression"

_cells, 2021, doi:10.3390/cells10030668_

Round 1
Reviewer 1 Report
In this manuscript, Causin et al. employ a literature-based review of the miRNAs associated with cervical cancer. While the review offers some insight in to how these molecule may play an important role in cervical cancer, the review needs to be simplified to avoid repetition and to make the overall message more clear.
- In Figure 1, Initial is spelt incorrectly in the heading.
- Line 67 - 'The mechanism of action of HPV with the host occurs in random regions of the human genome'. This sentence does not make sense. Do the authors mean integration of HPV within the host? This should be clarified.
- Line 72 - ', the mechanisms of action of miRNAs in the progression of cervical cancer uterus have been studied'. The word uterus should be removed.
- Line 75, 100 - no capitalisation of miRNA at beginning of sentence.
- Line 90 - 'Several factors are necessary for the development of CC, including mainly the interaction of viral and environmental factors'. Remove the word mainly.
- Line 95 - ref 28 seems inappropriate here as it is a review on miRNA, not epigenetic mechanisms.
- Lin 101 - '. In various cases of cancer, oncomiRs were found to be overexpressed or upregulated, and inversely, tsmiRs were downregulated [31]'. This statement is redundant due to the nature of oncomiRs and tsmiRs.
- Line 102-104 - this is not necessarily true - CC could be treated without considering the roles of miRNAs in CC progression.
- The authors show the identified miRNAs in both table form and in two different figures - this is unnecessary and should be reduced for brevity i.e. figures 4 and 5 show largely the same thing.
Author Response
Firstly we would like to thank you immensely for all your comments.
In this manuscript, Causin et al. employ a literature-based review of the miRNAs associated with cervical cancer. While the review offers some insight in to how these molecules may play an important role in cervical cancer, the review needs to be simplified to avoid repetition and to make the overall message clearer.
- In Figure 1, Initial is spelt incorrectly in the heading.
- Answer: Thank you very much for the correction, figure 1 has been adjusted, see figure 1 (file in attachment).
- Line 67 - 'The mechanism of action of HPV with the host occurs in random regions of the human genome'. This sentence does not make sense. Do the authors mean the integration of HPV within the host? This should be clarified.
- Answer: This paragraph has been rewritten see line 67.
- Line 72 - ', the mechanisms of action of miRNAs in the progression of cervical cancer uterus have been studied'. The word uterus should be removed.
- Answer: The word “uterus” has been removed, see line 74.
- Line 75, 100 - no capitalization of miRNA at beginning of a sentence.
- Answer: The words have been corrected, see lines 75 and 100.
- Line 90 - 'Several factors are necessary for the development of CC, including mainly the interaction of viral and environmental factors. Remove the word mainly.
- Answer: The word "mainly" has been removed, see line 90.
- Line 95 - ref 28 seems inappropriate here as it is a review on miRNA, not epigenetic mechanisms.
- Answer: The reference cited was exchanged to a new one (PMID: 33290823), please check line 95.
- Line 101 - '. In various cases of cancer, oncomiRs were found to be overexpressed or upregulated, and inversely, tsmiRs were downregulated [31]'. This statement is redundant due to the nature of oncomiRs and tsmiRs.
- Answer: We agree, that sentence has been removed.
- Line 102-104 - this is not necessarily true - CC could be treated without considering the roles of miRNAs in CC progression.
- Answer: This sentence has been modified. Here, in fact, we would like to explain that in the future these miRNAs can be used as therapeutic targets in personalized medicine for CC. For that, it is necessary to first understand the functions of these molecules in the progression of the CC, see lines 100 to 103.
- The authors show the identified miRNAs in both tables form and in two different figures - this is unnecessary and should be reduced for brevity i.e., figures 4 and 5 show largely the same thing
Answer: Answer: We would like to stick with both figures since figure 4 didactically show the downregulated and upregulated miRNAs in each of the cancer hallmarks. While figure 5 demonstrates the possible targets for each of these miRNAs and the possible pathways of the CC hallmarks.

Reviewer 2 Report
The paper by Causin et al. discusses the role of miRNAs in cervical cancer progression.
The authors performed a systematic review according to the PRISMA guideliness. Materials and methods sections are described very clearly. The manuscript summarizes the current knowledge about the role of miRNA in cancer progression. However, in my opinion, the study does not indicate the clinical significance of the results of systematic review and there are no further directions of research suggested in manuscript.
Minor points:
line 248 - upexpression word is not correct.
Several references are missing - I suggest the addition of recent review articles discussing cervical cancer pathogenesis and treatment (PMID 30638582) and the role of microRNA in cancer (PMID 33321819). Moreover, I suggest discussing the future of miRNA as therapeutics (PMID: 32660045, PMID: 31156715) and as diagnostic tool (PMID: 31979244)
Author Response
First, we would like to thank you immensely for all your comments.
The paper by Causin et al. discusses the role of miRNAs in cervical cancer progression.
The authors performed a systematic review according to the PRISMA guidelines. Materials and methods sections are described very clearly. The manuscript summarizes the current knowledge about the role of miRNA in cancer progression. However, in my opinion, the study does not indicate the clinical significance of the results of a systematic review and there are no further directions of research suggested in the manuscript.
Answer: The discussion was remodeled and added important points about the miRNAs as diagnostic biomarkers and prediction of therapy. In addition, we held a brief discussion on the therapeutic use of miRNAs in the CC.
- Line 248 - upexpression word is not correct.
- Answer: The word "upexpression" has been replaced by "overexpression".
- Several references are missing - I suggest the addition of recent review articles discussing cervical cancer pathogenesis and treatment (PMID 30638582) and the role of microRNA in cancer (PMID 33321819). Moreover, I suggest discussing the future of miRNA as therapeutics (PMID: 32660045, PMID: 31156715) and as a diagnostic tool (PMID: 31979244).
- Answer: The reference (PMID 30638582) was inserted in line 53, a new paragraph was added in the introduction of lines 268 to 279. The reference (PMID 33321819) was inserted in line 97, and the reference (PMID: 32660045, PMID: 31156715) was inserted in lines 281 to 291. A paragraph on the diagnostic use of miRNAs has been inserted, see lines 293 to 300.
Round 2
Reviewer 1 Report
The authors have made the vast majority of the changes I requested. However, I still believe that figure 4 and 5 should be combined to make the manuscript clear and easier for the reader to follow.